# *Drosophila melanogaster*: How and Why It Became a Model Organism

**DOI:** 10.3390/ijms26157485

**Published:** 2025-08-02

**Authors:** Maria Grazia Giansanti, Anna Frappaolo, Roberto Piergentili

**Affiliations:** Istituto di Biologia e Patologia Molecolari (IBPM) del Consiglio Nazionale delle Ricerche (CNR), at Dipartimento di Biologia e Biotecnologie, Università Sapienza di Roma, Piazzale Aldo Moro 5, 00185 Rome, Italy; mariagrazia.giansanti@cnr.it (M.G.G.); anna.frappaolo@cnr.it (A.F.)

**Keywords:** *Drosophila*, model organism, human diseases

## Abstract

*Drosophila melanogaster* is one of the most known and used organisms worldwide, not just to study general biology problems but above all for modeling complex human diseases. During the decades, it has become a central tool to understand the genetics of human disease, how mutations alter the behavior and health of cells, tissues, and organs, and more recently to test new compounds with a potential therapeutic use. But how did this small insect become so crucial in genetics? And how is it currently used in the study of human conditions affecting millions of people? In this review, we retrace the historical origins of its adoption in genetics laboratories and list all the advantages it provides to scientific research, both for its daily usage and for the fine tuning of gene regulation through genetic engineering approaches. We also provide some examples of how it is used to study human diseases such as cancer, neurological and infectious diseases, and its importance in drug discovery and testing.

## 1. Introduction: A Brief Historical Background

*Drosophila melanogaster*, commonly known as fruit fly or vinegar fly, is originally an African species, with all non-African lineages having a common origin [1]. The term “drosophila” means “dew-loving”: it is a modern scientific Latin adaptation from the Greek words δρόσος, drósos, “dew”, and φιλία, philía, “lover”. The term “melanogaster” means “black-belly”, and comes from the Greek words μέλας, mélas, “black”, and γᾰστήρ, gastḗr, “belly”. Its history as a model organism in biological research started around 1900–1901. According to Thomas Hunt Morgan’s biography [2], Charles William Woodworth, an American entomologist, was the first scientist to breed this insect in captivity at Harvard University, and he suggested to William Ernest Castle to use it in genetics studies, in parallel to mice, after Castle’s personal rediscovery of Gregor Mendel’s work in 1900. The studies of Castle and his group on inbreeding interested the entomologist Frank Eugene Lutz, who worked on the genetics of this insect at the Carnegie Institution’s new Station for Experimental Evolution at Cold Spring Harbor, Long Island, New York, from 1904 to 1909. Lutz introduced the fruit fly to T. H. Morgan, who was seeking less expensive organisms—compared to Castle’s mice lines—that could be bred in the very limited space of his laboratory. Starting in 1909, Morgan was soon able to identify and isolate many visible mutants and to determine the localization and behavior of genes; in January 1910, he discovered his first *Drosophila* mutant, a white-eyed male that he showed to be affected by a mutation on the X chromosome [3] (Figure 1A). Later, together with his group and especially with the help of his student Alfred Henry Sturtevant [4], he was also able to map and align other genes to chromosomes, creating one of the first genome-wide genetic maps (Figure 1B).

## 2. People Who Contributed to Making the Fruit Fly a Reference Model Organism

Morgan’s discoveries boosted the research on this organism, which soon became a reference for genetics studies. This is also witnessed by five Nobel Prizes in Physiology or Medicine awarded to drosophilists over time (the official statement being quoted below).

In 1933, T. H. Morgan was awarded “for his discoveries concerning the role played by the chromosome in heredity” (see above).

In 1946, Hermann Joseph Muller was awarded “for the discovery of the production of mutations by means of X-ray irradiation”. Muller collaborated with Morgan starting in 1910 and joined his group in 1912–1914; thanks to the balancer chromosome ClB, found in 1919, he was able to demonstrate (starting at the end of 1926) [7] a clear, quantitative connection between radiation administration and the induction of lethal mutations on the X chromosome [8,9].

In 1995, Edward Bok Lewis, Christiane Nüsslein-Volhard, and Eric Frank Wieschaus were awarded “for their discoveries concerning the genetic control of early embryonic development”. The genes involved in these phenomena were identified by producing random mutations in flies allowing, by verifying developmental defects, simple modifications or its total absence, to identify exactly which genes were affected by the induced mutations and to detect those specific and crucial for *Drosophila* development [10,11].

In 2011, Bruce Alan Beutler and Jules Alphonse Hoffmann were awarded half the prize “for their discoveries concerning the activation of innate immunity” and the other half of the prize was awarded to Ralph Marvin Steinman “for his discovery of the dendritic cell and its role in adaptive immunity”. Hoffmann found that the absence of an innate defense system in flies, caused by mutations in the *Toll* gene, died when infected with bacteria and fungi. Beutler identified that Toll-like receptors were also present in mice, showing the evolutionary conservation of this defense. Steinman discovered dendritic cells and their ability to activate T cells; therefore, the signal generated by the Toll-like receptors is perceived by the dendritic cells, which in turn activate the T cells, thus avoiding the destruction of the body’s own molecules [12,13,14].

In 2017, the Nobel Prize in Physiology or Medicine was awarded jointly to Jeffrey Connor Hall, Michael Rosbash, and Michael Warren Young “for their discoveries of molecular mechanisms controlling the circadian rhythm”. The circadian rhythm in an internal, biological clock which allows organisms to keep night/day rhythm. The scientists isolated independently (Hall and Rosbash at Brandeis University and Young at Rockefeller University) a gene (*period*, *per*) encoding a protein during the night, which is then degraded during the day, thus syncing flies with Earth’s rotation, identifying the so-called transcriptional translational feedback loop (TTFL) model for the generation of autonomous oscillator with a period of ∼24 h [15,16,17,18].

Other notable drosophilists exist, who made fundamental discoveries in biology. A limited and incomplete list follows; for detailed explanations on the techniques, we redirect the reader to specific reviews [19,20].
Walter Jakob Gehring discovered the homeobox in 1983 [21] and was also involved in the development and application of enhancer trapping methods.Michael Levine contributed to the discovery of the homeobox with his mentor Gehring and discovered the modular organization of the regulatory regions of developmental genes and their regulatory regions [22,23].Seymour Benzer used forward genetics to investigate the genetic basis of various behaviors such as phototaxis and learning and discovered the first circadian rhythm mutants in *Drosophila* [24,25,26,27]. Concerning this insect, he also studied neurodegeneration and aging and isolated the first long-life mutant called *Methuselah* [28].Gero Andreas Miesenböck is known as the founder of optogenetics [29,30].Gerald Mayer Rubin pioneered the use of transposable P elements in genetics and led the public project to sequence the *Drosophila* genome [31,32].Allan C. Spradling is considered a leading researcher in the developmental genetics of the fruit fly egg [33,34] and is renowned for his research on stem cells [35,36].Andrea Hilary Brand developed with Norbert Perrimon the GAL4/UAS system, “a fly geneticist’s Swiss army knife” [37]. She also worked on the biology of neural stem cells and the ability of neurons to regenerate after damage [38].Norbert Perrimon developed several techniques used in *Drosophila* genetic research (GAL4/*UAS* system, FLP-FRT DFS method, in vivo RNAi, CRISPR/Cas9) [37,39] and made substantial contributions to signal transduction and developmental biology and physiology [40].John R. Carlson studied insect chemosensation, particularly the olfactory and taste receptor genes [41,42].Leslie Birgit Vosshall studied the genetic and behavioral mechanisms involved in the olfaction and feeding behavior of fruit flies, mosquitoes, and humans [43].David Takayoshi Suzuki studied dominant temperature-sensitive (DTS) phenotypes and is a famous science broadcaster [44,45].Michael Ashburner identified a cascade of genetic controls in the post-larval development triggered by ecdysone (polytene chromosome puffs) [46]; he was also a member of the consortium involved in the sequencing and annotation of the *Drosophila* genome and actively participated in setting up the FlyBase, Gene Ontology, and ChEBI databases [47,48].


## 3. Why Use *Drosophila melanogaster*?

From the very beginning, the advantages of breeding fruit flies in captivity were evident and others were added over time [49,50] (Figure 2).

First, fruit flies are cost-effective, as they can be bred using a fruit-based, even hand-made, medium. As said, this was one of the reasons why Morgan started his studies on this organism. In addition, fly progeny is large—a female lays up to 100 eggs per day, and ca. 2000 in a lifetime, but occupies only a minimal space in the laboratory. Another advantage is their simple manipulation, as flies can be safely and readily anesthetized with ether, carbon dioxide gas, or by cooling, and handled using a brush or (like one century ago) a bird feather.

Formal genetics in fruit flies is one of the most advanced among model organisms: recessive lethal “balancer chromosomes” exist, inhibiting crossing-over and carrying visible genetic markers, which can be used to maintain mutations in heterozygosis across generations [51]. In addition, males do not show meiotic recombination, facilitating genetic studies. Additionally, phenotypic characterization in *Drosophila melanogaster* is quite straightforward: flies show ample visible phenotypic variation that has been extremely useful since the beginning of their use for the construction of genetic maps (Figure 1) and for planning selected crosses to study its genetics. This is also achieved thanks to the evident sexual dimorphism: males and females are readily distinguished, and virgin females can be easily identified either by time of pupal eclosing (ca. 8 h) or by phenotype (light-colored, translucent abdomen). The effects of mutations can be effortlessly studied over time, since the fruit fly life cycle is short, allowing researchers to obtain a new generation of insects within 10 to 15 days, depending on rearing temperature; moreover, their limited life span, up to two months, permits fast studying of aging phenomena.

As for cellular biology, *Drosophila* cytogenetics is easy too: fruit flies possess a simple karyotype, made of only eight chromosomes [51,52], and the presence of polytene chromosomes in salivary glands [53] provides simplified cytogenetics for the study of chromosome abnormalities (structural mutations) and function (puffs) [54,55]. In addition, the availability of a very large number of reagents allows one to easily visualize protein behavior by immunofluorescence to study fundamental aspects of life such as cell division and development.

At the molecular level, *Drosophila* genetic engineering has become ever more powerful over the decades, as also shown in the previous section: genetic transformation techniques have been available since the 1980s and the complete genome of *Drosophila melanogaster* was sequenced and first published 25 years ago [56]. Moreover, genes in the fly are evolutionary conserved: a study by the National Human Genome Research Institute comparing fruit fly and human genomes estimated that about 60% of genes are conserved between the two species and, of them, ca. 75% have a recognizable match of known human disease genes [57].

Last but not least, flies still undergo very few legal restrictions regarding their use, manipulation, and worldwide exchange among scientists, compared to mammals or other vertebrates, making scientific communication extremely powerful.

## 4. Using *Drosophila melanogaster* to Study Human Conditions

The above-mentioned conservation of human disease-associated genes between flies and humans [57] is one of the reasons why this model organism is widely used for the study of human pathologies in almost any field of medicine. Fruit flies offer additional advantages that are not present in other model organisms. The lower number of genes in the genome is associated with reduced genetic redundancy and simpler gene families. Moreover, the powerful and versatile genetic toolkit available in *Drosophila* permits the creation of mutants in multiple genes at the same time, allowing researchers to investigate protein interactions as well as to mimic the pleiotropic characteristics of several human diseases, such as cancer, obesity, sleep, aging, brain injuries, polycystic kidney disease, amyotrophic lateral sclerosis [58], and neurological/neuromuscular disorders, such as Parkinson’s disease [59].

### 4.1. Drosophila melanogaster as a Model in Cancer Research

Cancer development is a multistep process involving the sequential activation of oncogenic pathways and the loss of tumor suppressors [60]. Research on *Drosophila melanogaster* has significantly contributed to identifying the genetics and the pathways that play oncogenic and tumor suppressor roles and to understanding the cellular mechanisms that drive tumor growth and invasion [61,62,63,64].

The first mutation associated with the hyperproliferation of *Drosophila* larval tissues was a recessive allele of *lethal giant larvae* (*lgl*), a gene identified in the 1930s by Bridges and reported by Hadorn in 1938 [65]. Mutant larvae carrying *lgl* mutations displayed disorganization and abnormal overproliferation in the brain, imaginal disks, and hematopoietic organs [66]. It was later demonstrated that *lgl* and other genes, which also include *scribble* (*scrib*) and *discs large* (*dlg*), control apicobasal polarity in epithelial cells and prevent neoplastic overgrowth in the larval brain and imaginal disks [67,68]. Importantly, consistent with the loss of cell polarity observed in 80% of human cancers, the expression levels of the human homologs of *scrib* and *dlg* are reduced in various cancer types [62,69].

Developmental and cell biology studies have revealed that most human tumor suppressors and oncogenes share conserved functions in *Drosophila* [62,63,70,71]. Combined with the superior genetic toolkit available in flies, this enables the modeling of specific genetic alterations commonly found in human patients and testing the drug’s therapeutic efficacy tailored to specific genotypes [64]. Beyond the expected similarities due to conserved genes misexpression, *Drosophila* were also revealed to be useful in studying tumor microenvironments and how cancer cells interact with each other, as well as with the healthy, surrounding cells [72]. Furthermore, *Drosophila* also emerged as useful in the study of cachexia, a multifactorial syndrome associated (among other conditions) with cancer and characterized by a deep energy imbalance causing significant weight loss mainly due to the loss of muscle and fat mass in the body [73]. Several cancer types have been modeled in *Drosophila* including colorectal, lung, and brain cancers [74].

#### 4.1.1. *Drosophila* Model for Colorectal Cancer

Colorectal cancer (CRC) is the second most fatal cancer type with 1.9 million new cases and 935,000 deaths reported worldwide in 2020 [75]. CRC development results from the activation of *RAS* oncogenes (*NRAS*/*KRAS*/*HRAS*) and/or the inactivation of tumor suppressors (*APC*, *SMAD4*, *TP53*, *LLGL1*) [76]. Genetically engineered mouse models (GEMMs) for intestinal tumors have greatly contributed to the discovery of molecular pathways underlying CRC development and invasion [77,78]. However, both the generation and the maintenance of CRC mice models are very challenging. Besides the genetic similarities between fruit flies and mammals, *Drosophila* CRC models display similar features of human CRC including altered cell differentiation and cell growth associated with the disruption of intestinal homeostasis [79]. Moreover, the *Drosophila* midgut and the mammalian intestinal tract share key functions and molecular characteristics, providing a well-suited cell system for modeling CRC [80]. Indeed, the intestines in both *Drosophila* and humans are maintained by proliferative cells that generate post-mitotic secretory cells and absorptive enterocytes [80]. To model CRC genotypes in the *Drosophila* hindgut, Bangi and coauthors used the patient genomic data from The Cancer Genome Atlas (TCGA) to generate 32 multigenic models which recapitulate key cancer pathologies, including cellular proliferation, basement membrane disruption, epithelial–mesenchymal transition, and distant metastasis [81]. The combination of oncogenic mutation *Ras^G12V^* with the knockdown of the tumor suppressors *p53*, *Pten*, and *apc* caused the most severe phenotypes. The study from Bangi and coauthors also explored the relationship between the genetic complexity and the tumor model’s response to specific FDA-approved cancer drugs. Importantly, the resistance to a panel of cancer drugs was correlated with multigenic combinations such as *Ras*, *p53*, *Pten*, and *apc*. Further exploring the mechanisms of drug resistance, the researchers demonstrated that the resistance to the PI3K/mTOR inhibitor BEZ235 was associated with *Ras* activation and the simultaneous *Pten* loss. However, the spread of CRC was successfully reduced through a two-step process when the CRC *Drosophila* models were treated with the AKT-activating compound SC79 followed by BEZ235. Importantly, this two-step approach was also effective in cultured human tumor cells, xenografts, and CRC GEMM, indicating that the CRC *Drosophila* models can be a useful platform for rapid and large-scale functional exploration of novel targeted therapies tailored to specific patients’ genotypes [81].

#### 4.1.2. *Drosophila* Model for Lung Cancer

Statistically, lung cancer is the leading cause of death among cancer diseases worldwide with non-small cancer cell lung cancer (NSCLC) accounting for approximately 85% of all diagnosed cases [75,82]. Conventional NSCLC treatments commonly include chemoradiotherapy combined with complementary targeted therapies. However, the current drugs, targeting lung cancer, have been proven to have limited success in tumor suppression due to emergent resistance and significant toxicity. Modeling lung cancer in *Drosophila* exploits the similarities between *Drosophila* tracheal and vertebrate lung development [83]. Similarly to the vertebrate lung, the *Drosophila* tracheal system consists of large primary tubes that branch into diminishing diameter segments and end in terminal sectors [83]. During *Drosophila* tracheal development, the gene *branchless* (*bnl*) is a key determinant of the tracheal branching pattern and functions as a ligand for the *breathless* receptor tyrosine kinase, the FGF (fibroblast growth factor) receptor homolog, expressed on developing tracheal cells [84]. FGF signaling is also involved in the process of vertebrate lung branching [85]. Levine and Cagan generated a *Drosophila* lung cancer model by tracheal targeted expression of oncogenic mutation *Ras1^G12V^* in combination with the knockdown of the PI3K-negative regulator *PTEN* [86]. Combining the targeted activation of Ras1 with the loss of PTEN activity led to early larval stage death, overproliferation of larval tissue, tracheal cell proliferation, and loss of pupal air sacs. The chemical screening of a library of 1192 FDA-approved drugs for the ability to improve animal survival identified a synergistic relationship between the MEK1/2 kinases inhibitor drug trametinib and the HMG-CoA reductase inhibitor fluvastatin. These two compounds rescued the tracheal cell proliferation defects, reduced whole-organism toxicity, and curtailed the growth of the *Ras1^G12V^*-positive A549 lung adenocarcinoma cell line [86]. *Drosophila* has also contributed to the development of novel therapeutic strategies for targeting *KIF5B-RET* fusions, which account for approximately 2% of all NSCLC patients [87]. The study of Das and Cagan [87] demonstrated that the C-terminal RET kinase domain of KIF5B-RET activates canonical signaling pathways, while its N-terminal KIF5B domain activates multiple receptor tyrosine kinases (RTKs), including epidermal growth factor receptor (EGFR) and fibroblast growth factor receptor (FGFR) signaling. The administration of drugs designed to inhibit RET alone proved ineffective in *KIF5B-RET*-transformed cells. However, combining the RET inhibitor with the EGFR inhibitor sorafenib or the microtubule inhibitor paclitaxel had strong efficacy in both *Drosophila* and human cell line *KIF5B-RET* models. While these therapies are currently awaiting validation in patients, the study on *KIF5B-RET* models suggests that novel therapeutic strategies would benefit from the use of *Drosophila*.

#### 4.1.3. *Drosophila* Model for Glioblastoma Multiforme

Gliomas represent the most frequent type of central nervous system (CNS) malignant primary tumors, with glioblastoma multiforme (GBM) having the highest mortality rate [88]. The most frequent genetic lesions in GBM and high-grade gliomas include the mutation or amplification of the epidermal growth factor receptor (EGFR) tyrosine kinase coding genes. Glioma-associated *EGFR* mutant forms display constitutive kinase activity, which drive cellular proliferation and migration by chronically stimulating Ras signaling [89]. Other common genetic lesions in GBM include loss of lipid phosphatase PTEN, which antagonizes the phosphatidylinositol-3 kinase (PI3K) signaling pathway, activating mutations in the p110α catalytic subunit of PI3K and constitutively active Akt [90,91]. To investigate the genetic basis of this disease, Read and coauthors developed a *Drosophila* glioma model [92]. They demonstrated that the constitutive coactivation of EGFR-Ras and PI3K signaling in *Drosophila* glia and glial precursors leads to neoplastic glial proliferation and transplantable tumor-like growths, which recapitulate the characteristics of human GBM and animal glioma models [92]. In turn, the cancer phenotypes observed in the *Drosophila* glioma model were found to rely on crucial downstream effectors of EGFR and PI3K signaling including the Tor, Myc, G1 Cyclins-Cdks, and Rb-E2F pathways [92]. Thus, *Drosophila* offers a valuable model system for identifying new therapeutic targets in GBM using genetic approaches.

Below is a summary of the use of the fruit fly in modeling the main cancers previously described (Table 1).

### 4.2. Drosophila melanogaster as a Model for Neurodegenerative and Neurodevelopmental Diseases

Brain disorders, including neurodegenerative and neurodevelopmental disorders, affect a significant portion of the human population, which has been estimated to be 43% of the global population [93,94]. Neurodegenerative diseases are characterized by the slow and progressive loss of neurons in discrete areas of the central nervous system, leading to motor and cognitive impairment [95]. Neurodevelopmental disorders are a group of disorders affecting brain development and function causing the inability to reach cognitive, emotional, and motor developmental milestones [96]. Neurodevelopmental disorders are associated with autism spectrum disorder (ASD), intellectual disability (ID), and epilepsy [96]. The fruit fly has emerged as a very good model organism to study the molecular mechanisms underlying many human neurological disorders due to several important characteristics. Cellular and molecular pathways, including membrane excitability, synaptic plasticity, and neuronal signaling, are highly conserved in *Drosophila* [97]. Moreover, fruit flies have a complex and compact CNS, protected by a blood–brain barrier, with neurons and glia, whose functions are very similar to those in vertebrates [98,99]. Although less complex, the *Drosophila* brain contains the counterparts of some mammal structures [98] such as the mushroom body, which modulates complex behaviors and corresponds to the human hippocampus [100]. Remarkably well-established assays have been developed in *Drosophila* to assess neurological functions such as hearing, flight, learning, memory, and circadian rhythmicity, as well as several behavioral assays [101,102,103,104,105]. Among the neurodegenerative diseases modeled in *Drosophila*, there are Alzheimer’s disease, Parkinson’s disease, Lewy Body Dementias, Frontotemporal dementia, Amyotrophic Lateral Sclerosis (ALS; also known as Lou Gehrig’s disease), Huntington’s Disease, Ataxia Telangiectasia, and neurodegeneration related to mitochondrial dysfunction or traumatic brain injury [64,94,98,99,106,107,108]. *Drosophila* models have also been generated for several neurodevelopmental diseases including Fragile-X syndrome, Neurofibromatosis Type 1, Rett syndrome, and Angelman syndrome [94,109,110,111]. Finally, addiction and sleep disorders may also be modeled in this organism [112]. Thus, investigating *Drosophila* models of neurodegeneration may shed light onto the identification of causative mutations in patients, the isolation of possible targets, and the exploration of new therapeutic approaches. Numerous reviews are available in the literature that describe in detail the major neurodegenerative and neurodevelopmental diseases listed above [64,94,98,99,106,107,108,109,110,111]. Thus, here we will focus only on recent findings in the study of PACS1-PACS2-WDR37 rare neurodevelopmental disorders, highlighting the advances achieved using fly models.

PACS (phosphofurin acidic cluster sorting) proteins are multifunctional membrane trafficking regulators that control cellular homeostasis and prevent disease states [113,114,115]. In humans, PACS proteins have been associated with neurodevelopmental disorders (NDDs), characterized by epileptic seizures, neurodevelopmental delay, and congenital abnormalities [116,117]. In vertebrates, the PACS family includes two *PACS* paralogous, *PACS1* and *PACS2* [115,118]. Schuurs–Hoeijmakers syndrome (SHMS, MIM #615009), also known as PACS1-NDD, is a rare autosomal dominant disorder characterized by ID and ASD, associated with epileptic seizures, hypotonia, feeding difficulties, distinctive abnormal craniofacial features, microcephaly, cryptorchidism in males, and congenital heart malformations [119]. The human PACS1 protein contains a Furin-binding region that interacts with cargo proteins and sorting adaptors containing specific phosphorylated acidic cluster motifs and regulates endosome-to-TGN (trans-Golgi network) trafficking [114,115]. In humans, individuals carrying a missense mutation in *PACS2* are affected by developmental and epileptic encephalopathies (Developmental and Epileptic Encephalopathy 66, MIM #618067) with neonatal/early-infantile onset and associated with extra-neurological features including autism, cerebellar dysgenesis, and facial dysmorphism [117]. Similarly to PACS1, PACS2 plays a role in the regulation of protein sorting. PACS2 has been involved in the trafficking from Golgi to endoplasmic reticulum (ER) and from endosomes to the TGN or plasma membrane [115,118,120,121].

In vertebrates, functional analysis of PACS proteins is complicated by the presence of two paralogs which can compensate for the loss of each other [122]. Conversely, *Drosophila melanogaster* harbors a single *PACS* gene (*dPACS*; [115,118]). Consistent with its well-known role in membrane trafficking, the dPACS protein is enriched at the Golgi stacks in spermatocytes [123]. The localization of the dPACS protein to Golgi membranes requires the wild-type function of two Golgi trafficking regulators, namely the conserved oligomeric Golgi (COG) complex subunit Cog7 and the phosphatidylinositol 4-phosphate [PI(4)P] effector GOLPH3 (Golgi phosphoprotein 3) [123]. The dPACS protein colocalizes with GOLPH3 and Cog7 at the Golgi stacks and is required for maintaining the Golgi architecture. dPACS is also essential for cell division in both dividing spermatocytes and neuroblasts. During cytokinesis, the loss of dPACS impairs central spindle stability and actomyosin ring constriction in dividing spermatocytes [123]. dPACS and GOLPH3 proteins form a complex and are mutually required for localization to the cleavage site. Thus, dPACS, by associating with GOLPH3, controls membrane trafficking that supports furrow ingression during cytokinesis. Furthermore, dPACS deficiency causes defects in tubulin acetylation and severe bang sensitivity, a phenotype associated with seizures in flies [123].

PACS1-NDD and PACS2 syndromes share strong similarity with the recently described WDR37 syndrome (neurooculocardiogenitourinary syndrome, MIM #618652), a neurological disorder characterized by epilepsy, colobomas, facial dysmorphology, cerebellar hypoplasia, developmental delay, and intellectual disability [124]. Importantly, flies carrying a null allele of *wdr37*, the fly orthologue of human *WDR37* (*WD40 repeat-containing protein*), exhibit the same bang-sensitive phenotype observed in dPACS mutants [124]. Indeed, WDR37 is a member of the WD40 repeat protein family reported as a PACS protein interactor in several studies [125,126,127]. Furthermore, it has been recently reported that the PACS1 and WDR37 proteins form a complex that is involved in the regulation of ER calcium homeostasis [127]. These data may underlie the overlapping clinical manifestations in PACS- and WDR37-deficient patients.

WDR37 is expressed ubiquitously [128], whereas PACS-1 and PACS-2 are broadly expressed in all tissues, with high levels in the brain [129]. During development, PACS1 protein levels are high in the embryonic brain and then decrease, suggesting a crucial role during the stages of neurodevelopment [130]. However, how these proteins can function in brain development and overall and the molecular mechanisms underlying PACS/WDR37 syndromes are still unknown. Accordingly, to date, no therapy is available beyond the treatment of symptoms through multidisciplinary approaches [131]. In this context, *Drosophila* in vivo models can be exploited for these purposes. The presence of a unique homolog of PACS proteins will offer great advantages to link the mechanisms of cellular trafficking and protein sorting with the pathological manifestation of PACS-related syndromes. Together, these findings suggest that *Drosophila* PACS/WDR37 disease models may significantly advance our knowledge of the molecular mechanisms underpinning human syndromes and lead to new therapeutic strategies.

Below is a summary of the use of the fruit fly in modeling rare neurodevelopmental disorders caused by PACS1-PACS2-WDR37 (Table 2).

### 4.3. Drosophila as a Model for Other Human Pathologies

Beyond cancer and neurodegeneration, *Drosophila* has been extensively used to study additional human conditions, even in cases where an evident similarity with the human body is lacking. For example, the fruit fly is a model for hepatic diseases, despite lacking a liver, since its fat bodies fulfill comparable functions with its human counterpart, such as energy storage, metabolism, and immune response. This allowed for investigating conditions such as obesity, insulin resistance, and non-alcoholic fatty liver disease caused either by genetic mutations or by exposure to environmental insults. *Drosophila* also helped in characterizing cardiac and muscular diseases, such as Duchenne muscular dystrophy and congenital myopathies [64].

Aging is another field in which *Drosophila* is a very valuable tool. Aging is a complex decline of an organism occurring over time and involves the degeneration of several bodily activities including metabolism, behavior, stress resistance, reproductive capacity, nervous system function, and immune capacity; this decline eventually results in death. The study of these phenomena in the fly is advantageous not only for its limited life span and conserved genes but also because *Drosophila* genomic resources are accessible to the public and the existing information on aging in the fly is readily available. All this coupled with the powerful molecular tools described above allow for either up- or down-regulating gene expression in specific tissues/organs or in a limited time frame [132]. Notably, a review on transgenerational inheritance and aging was published in 2018 [133].

### 4.4. Drosophila as a Model for Human Infectious Diseases

Thanks to its well characterized immune response, *Drosophila* has been repeatedly used for the study of viruses threatening human health [134]. Among others, we report in Table 3 some of the most meaningful contributions of *Drosophila* in their characterization.

### 4.5. Drosophila as a Model for Drug Identification and Testing

A typical pipeline in disease treatment is represented by a few steps, of which drug discovery is the initial one, followed by preclinical screening and development, clinical trials, drug approval, and commercialization. In this pipeline, *Drosophila* may play a pivotal role in the preclinical phase, characterized by in vitro and in vivo screening, pharmacokinetics, pharmacodynamics, and toxicity evaluation, thanks to the fly/human similarities not only at the genetic but also at the physiological level. In this context, the fruit fly is extremely useful as a pathway discovery platform, allowing researchers to identify how (conserved) key proteins interact with the molecule of interest. For these reasons, several studies involve this organism in the early phases of drug testing. Studies are numerous and involve cancer (reviewed in [74,157]), infectious diseases (reviewed in [158]), neurological and neuromuscular diseases [159,160,161]. An important advantage of using the fly is to perform high-throughput genetic and pharmacological screenings of libraries of small molecules potentially targeting key proteins. Another use of this model is for testing the toxicity of drugs, allowing researchers to highlight both the safety and potential side effects of newly identified drugs as well as environmental toxins and pollutants, such as heavy metals or pesticides [64]. These studies allow researchers not only to discover any effect of these substances but also to explore the mechanisms by which they act and, consequently, plan modifications in their chemistry or administration [62].

### 4.6. Limits of Drosophila melanogaster When Modeling Human Diseases

Despite the many advantages discussed before, we should keep in mind that the fruit fly is an insect, thus evolutionarily quite distant from vertebrates, in general, and *H. sapiens*, in particular. Consequently, its body organization shows evident differences from that of vertebrates, and in many cases its simplified genome—which is central for the analysis of complex traits with limited genome manipulation—might hide the complex network of controls and interactions present in human cells, thus potentially revealing only a limited number of features connected to human disease. These aspects have been thoroughly discussed in the literature [162,163,164,165]. Here, we will only summarize the main features of this topic.

Cancer is a multifactorial disease involving tissue-specific microenvironments, immune interactions, and metabolic reprogramming. While *Drosophila* has been widely used to study oncogenes and tumor suppressors, this model organism has some limits in human cancer biology. For instance, *Drosophila* does not possess an adaptive immune system, which is crucial for tumor surveillance and immunotherapy responses [166,167]. Furthermore, the flies’ epithelial tissues do not undergo metastasis in the same way as human cancers. Although invasive behavior can be induced in fly models, the absence of a vascular system and lymphatic drainage precludes the modeling of hematogenous spread and tumor angiogenesis [64,168].

*Drosophila* has been instrumental in elucidating basic mechanisms of neurodevelopment and neurodegeneration. However, its nervous system is significantly simpler than that of humans. For example, while *Drosophila* models expressing human amyloid precursor protein (APP) and tau have been used to study Alzheimer’s disease, the fly lacks a cerebral cortex and hippocampus, which are critical for human cognition and memory, and cannot replicate the full spectrum of cognitive and behavioral symptoms seen in humans [168,169]. Psychiatric disorders such as schizophrenia and autism spectrum disorders (ASDs) involve complex behaviors and brain circuitry, which are absent in flies. Although some ASD-related genes like *FMR1* have orthologs in *Drosophila*, the behavioral phenotypes are rudimentary when compared to humans. Moreover, while Parkinson’s disease has been modeled using alpha-synuclein expression, the fly lacks dopaminergic neuronal subtypes and the glial diversity found in humans [165,168].

Diseases such as diabetes, obesity, and cardiovascular disorders are influenced by complex interactions between genetics, environment, and lifestyle. While *Drosophila* has insulin-like peptides and can model aspects of metabolic regulation, it lacks key organs such as the pancreas, liver, and kidneys. This limits its utility in studying diabetic complications like nephropathy and retinopathy. Cardiovascular diseases are also difficult to model due to the fly’s open circulatory system and the lack of a heart with chambers and coronary circulation. Moreover, chronic inflammation and immune dysregulation, central to many multifactorial diseases, are not adequately represented in the fly’s innate immune system [165,170].

The different fly/human immune system also limits the use of *Drosophila* to study innate immune responses and host–pathogen interactions. *Drosophila*’s innate immune system shares some pathways with mammals, such as the Toll pathway, but it lacks adaptive components such as T and B cells, immunoglobulins, and major histocompatibility complex (MHC) molecules. This makes *Drosophila* poorly suitable for modeling diseases that involve adaptive immunity, such as HIV/AIDS, tuberculosis, and many viral infections. Studies have also shown that *Drosophila* cannot reliably distinguish between virulent and non-virulent Gram-positive bacteria when infected via injection, highlighting its limitations as a host model for bacterial pathogenesis [171]. Additionally, the fly’s microbiome and gut physiology differ significantly from humans, limiting its relevance in studying gastrointestinal infections and microbiota-related diseases [64].

Also in the field of the study of aging, in which *Drosophila* has emerged as a valuable model, the fly shows some limits. Aging in humans involves complex tissue-specific degeneration, immune senescence, and chronic diseases that are not fully recapitulated in flies. For example, while oxidative stress and insulin signaling pathways are conserved, the absence of organs like the kidney and liver limits the study of age-related diseases such as chronic kidney disease and hepatic fibrosis. Furthermore, the fly’s simplistic physiology and lack of regenerative capacity in adult tissues constrain its utility in modeling human aging [172].

As for drug discovery and testing, *Drosophila* has been used for the high-throughput screening of drug candidates, especially in genetic modifier screens. However, its utility in pharmacokinetics and pharmacodynamics is limited. The fly’s absorption, distribution, metabolism, and excretion (ADME) processes differ significantly from humans. For instance, *Drosophila* lacks the cytochrome P450 enzyme diversity found in human liver, which is crucial for drug metabolism. This makes it difficult to assess drug–drug interactions and metabolic stability. Moreover, the different blood–brain barrier and absence of human-like organ systems means that tissue-specific drug effects cannot be reliably studied. Unsurprisingly, regulatory agencies do not accept *Drosophila* data for preclinical safety evaluation, necessitating validation in mammalian models [62,165,168]. Table 4 summarizes the main fly/human differences in drug metabolism.

## 5. Conclusions

*Drosophila melanogaster* is intimately linked to the history and development of genetics. Its use to investigate the inheritance of characters just after Mendel’s rediscovery, coupled with its ease of use and low cost of rearing prompted many scientists to adopt it as a tool to explore the newly born discipline. Over the years, its importance grew steadily, especially thanks to the work of TH Morgan and his staff, including Alfred Sturtevant, who elaborated the first genetic map; Calvin Bridges, who helped understand the role of chromosomes in heredity genes; and Hermann Muller, who discovered the role of X-rays in mutagenesis. The elaboration of new molecular tools to manipulate its DNA, together with the simplified yet evolutionary conserved genome, further boosted its importance in genetics, making it one of the most used animal models worldwide. Today, after more than one century, the fruit fly still provides invaluable clues into understanding human diseases and helps elaborate on new therapeutic strategies and testing new chemical compounds with an immense impact on human health.

Despite its strengths in genetic manipulation and pathway discovery, *Drosophila melanogaster* has inherent limitations in modeling human diseases, particularly those involving complex organ systems, immune responses, and multifactorial etiologies (Figure 3). Its role in drug discovery is primarily in early-stage screening, with limited translational value for pharmacological testing. Therefore, while *Drosophila* still remains a valuable tool in biomedical research, its findings must be complemented with data from more physiologically relevant models, especially mammalian ones (e.g., rats, mice, pigs, monkeys).

## Figures and Tables

**Figure 1 ijms-26-07485-f001:**
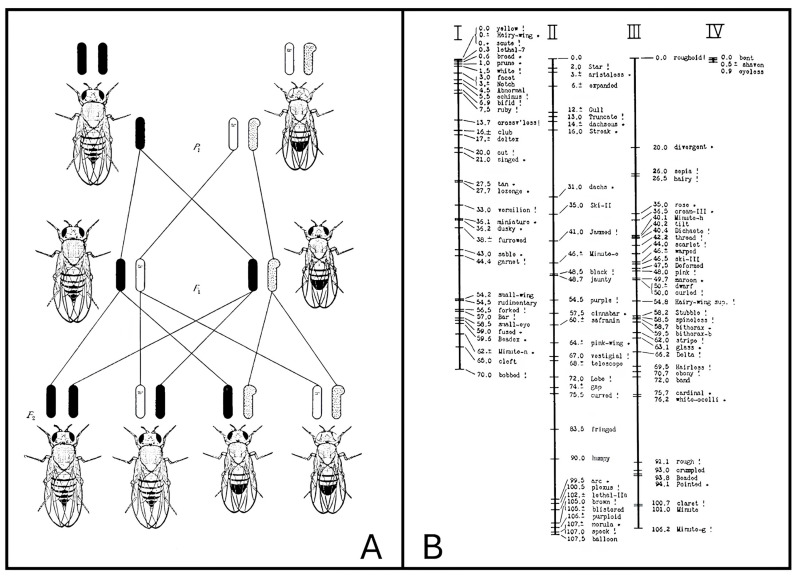
Two of the main discoveries by Morgan and his group. (**A**) Inheritance of the *white* gene, mapping on the X chromosome. (**B**) Linkage maps of *Drosophila melanogaster* visible variations; chromosome “I” is the X chromosome. Both pictures come from the original Morgan’s book published in 1926 [5]. (Copyright note: images can be freely used because all copyrightable works published in the United States before 1930 are now in the public domain due to copyright expiration [6].)

**Figure 2 ijms-26-07485-f002:**
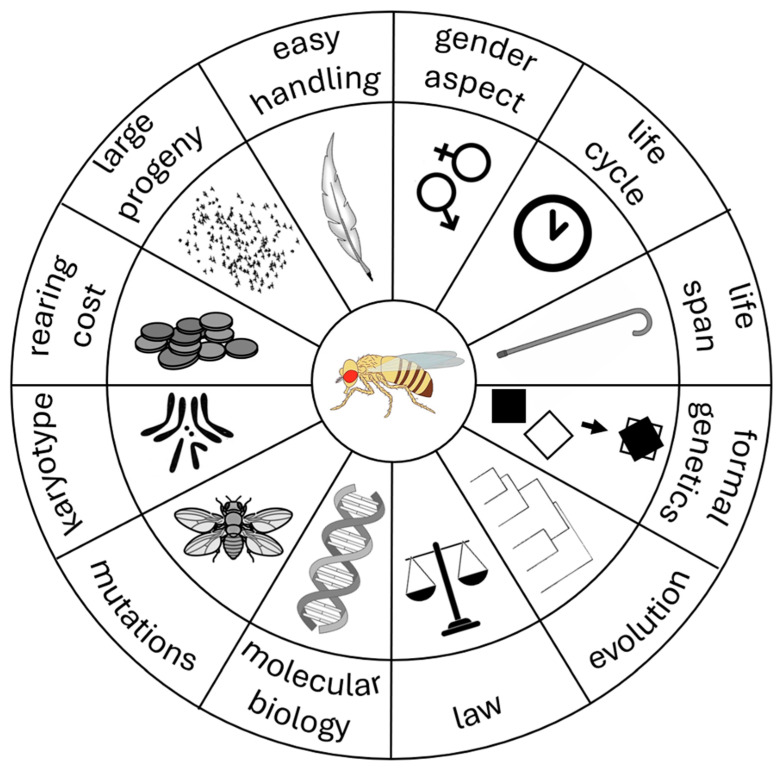
The advantages of studying *Drosophila melanogaster*.

**Figure 3 ijms-26-07485-f003:**
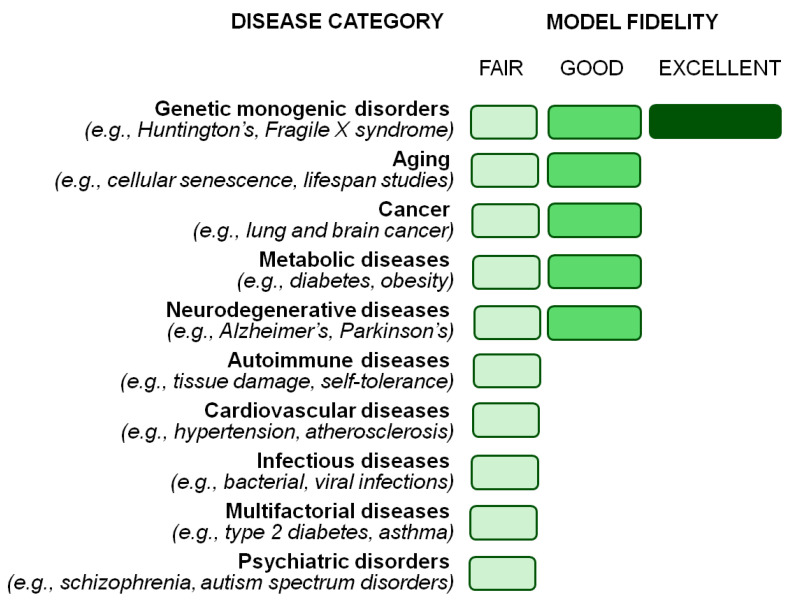
Main disease categories for which *Drosophila* is a valuable model, and the relative model strength. Data are listed top-down by (i) strength and (ii) alphabetical order.

**Table 1 ijms-26-07485-t001:** Using *Drosophila melanogaster* to model human cancers.

Cancer Type	Genetic Mutations	Why *Drosophila*	Ref.
Colorectal cancer	*Ras^G12V^*, *p53*, *Pten apc*	Similarly to human CRC, *Drosophila* CRC models display altered cell differentiation and cell growth associated with the disruption of intestinal homeostasis	[81]
Lung cancer	*Ras^G12V^ Pten*	Similarities between the *Drosophila* tracheal and the vertebrate lung development Tracheal cell proliferation and loss of pupal air sacs	[86]
Lung cancer	*KIF5B-RET*	*Drosophila KIF5B-RET* model suggests novel therapeutic strategies for targeting *KIF5B-RET* fusions	[87]
Glioblastoma multiforme	Glial specific expression of activated EGFR and *dp110* (*repo>dEGFR^λ^*; *dp110^CAAX^*)	Neoplastic glial proliferation and transplantable tumor-like growths The cancer phenotypes rely on crucial downstream effectors of EGFR and PI3K	[92]

**Table 2 ijms-26-07485-t002:** Using *Drosophila melanogaster* to investigate rare neurodevelopmental disorders caused by PACS1-PACS2-WDR37.

NDD/#MIM	Human/*Drosophila* Gene	The *Drosophila* Disease Model	Ref.
PACS1-NDD MIM#615009	*PACS1*/*dPACS*	Loss of dPACS leads to defects in tubulin acetylation and severe bang sensitivity, a phenotype associated with seizures	[123]
PACS2-NDD MIM#618067	*PACS2*/*dPACS*	Loss of dPACS leads to defects in tubulin acetylation and severe bang sensitivity, a phenotype associated with seizures	[123]
WDR37-NDD MIM #618652	*wdr37*	Loss of *wdr37* causes bang sensitivity and a defect in grip strength	[124]

**Table 3 ijms-26-07485-t003:** Using *Drosophila melanogaster* to investigate the biology of viruses threatening human health.

Virus Full Name	Virus Acronym	Associated Disease	Why *Drosophila*	Ref.
Human immune-deficiency virus 1	HIV1	Acquired immune deficiency syndrome (AIDS)	it allowed us to understand the role of Toll and JNK pathways during infection	[135,136]
Dengue virus	DENV	Dengue hemorrhagic fever (DHF)	it allowed for a better understanding of the role of RNA interference (RNAi) in infection control	[137,138]
Severe acute respiratory syndrome coronavirus	SARS-CoV	Severe acute respiratory syndrome (SARS)	it allowed for better understanding the protein–protein interactions between viral and host proteins	[139,140]
Sindbis virus	SINV	Sindbis fever	it allowed for better understanding the entry mechanism of the virus inside cells and the role of the ERK pathway in *Drosophila* and mosquito (the natural virus carrier) intestinal immunity	[141,142]
West Nile virus	WNV	West Nile fever	it allowed for exploring the possibility to control the infection via RNAi	[143,144]
Influenza A virus	IAV	Pandemic flu	it allowed for identifying several conserved host factors important for virus replication	[145]
Vesicular stomatitis virus	VSV	Oncolytic virus causing a flu-like condition	it allowed for studying of the role of Toll-7 in controlling virus infection	[146]
Epstein–Barr virus	EBV	Mononucleosis; also involved in cancer and multiple sclerosis	it allowed for the identification of human EBV-targeted tumor suppressors	[147,148]
Human cytomegalovirus	HCMV	Birth defects	it provided a model to study how HCMV impairs embryonic development	[149]
Simian virus 40	SV40	Debated role in oncogenesis	possible oncogenetic routes have been disclosed in the fly	[150]
Vaccinia virus	VACV	Rash and fever; also used as a vaccine for smallpox	it allowed for the identification of host factors required for viral entry	[151]
Severe acute respiratory syndrome coronavirus 2	SARS-CoV-2	Coronavirus disease (COVID)-19	it allowed for the identification of key functional interactions between viral factors and host proteins and its relationship with cardiovascular and neuromuscular complications in humans	[152,153,154,155,156]

**Table 4 ijms-26-07485-t004:** A comparative overview of the drug metabolism pathways between *Drosophila melanogaster* and *Homo sapiens*, highlighting key differences in organ systems, enzyme families, and pharmacokinetic processes relevant to drug absorption, distribution, metabolism, and excretion (ADME), as well as the presence of a different blood–brain barrier (BBB).

Category	*Drosophila melanogaster*	*Homo sapiens*
Organs Involved	Fat body, midgut, Malpighian tubules (analogous to liver and kidney functions)	Liver, kidneys, intestines, lungs
Enzyme Systems	Cytochrome P450 monooxygenases (less diverse), esterases, glutathione S-transferases	Extensive cytochrome P450 families (CYP1, CYP2, CYP3), UGTs, SULTs, esterases
Blood–Brain Barrier	Present but structurally and functionally simpler; lacks tight junctions of vertebrate BBB	Complex structure with tight junctions, astrocytic end-feet, and selective permeability
Absorption	Primarily through ingestion; limited oral bioavailability studies	Oral, intravenous, subcutaneous, transdermal, etc.
Distribution	Open circulatory system; hemolymph distributes compounds	Closed circulatory system; plasma protein binding and tissue perfusion
Metabolism	Simplified metabolic pathways; limited phase I and II reactions	Complex phase I (oxidation, reduction, hydrolysis) and phase II (conjugation) metabolism
Excretion	Malpighian tubules and hindgut; excretion into feces	Renal (urine), biliary (feces), pulmonary, and sweat excretion routes

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
