# Peer review of "Drosophila melanogaster*: How and Why It Became a Model Organism"

_ijms, 2025, doi:10.3390/ijms26157485_

Round 1
Reviewer 1 Report
Comments and Suggestions for Authors
The authors wrote an interesting review on the impact and the importance of Drosophila as a model organism for the study of human diseases. The manuscript is well written, scientifically relevant and all the concepts are clearly explained. No modifications are needed for the publication.
Author Response
Comment 1: The authors wrote an interesting review on the impact and the importance of Drosophila as a model organism for the study of human diseases. The manuscript is well written, scientifically relevant and all the concepts are clearly explained. No modifications are needed for the publication.
Response 1: We are very grateful to Reviewer 1 for the very positive comments, we are very happy that s/he enjoyed our manuscript.
Reviewer 2 Report
Comments and Suggestions for Authors
Giansanti et al. present a precise and up-to-date review, comprehensively detailing the advantages of using Drosophila as a model organism. The authors meticulously describe the organism's biology and genetics and provide a thorough overview of significant scientific discoveries associated with it. Furthermore, the review effectively surveys a range of human pathologies that can be modeled in Drosophila.
While the review is exceptionally well-executed, I have a few minor suggestions to enhance its pedagogical value and comprehensiveness:
- I recommend adding a paragraph that summarizes the limitations of Drosophila as a model organism. This would ensure the review is 100% didactically robust and all-encompassing.
- It would be beneficial to include a table summarizing the various human pathologies described as potentially modellable in Drosophila. This would improve the readability and utility of this section.
- Finally, I suggest adding a few more references to the list of valuable discoveries proposed on page 3 (e.g., those from Levine, Brand, Perrimon, Rubin, etc.) to further solidify this section.
Beyond these minor suggestions, I find the review to be complete and well-written. I approve its publication in IJMS.
Author Response
Comment 1: Giansanti et al. present a precise and up-to-date review, comprehensively detailing the advantages of using Drosophila as a model organism. The authors meticulously describe the organism's biology and genetics and provide a thorough overview of significant scientific discoveries associated with it. Furthermore, the review effectively surveys a range of human pathologies that can be modeled in Drosophila.
Response 1: We are very grateful to Reviewer 2 for the very positive comments, we are very happy that s/he enjoyed our manuscript, and we hope that our responses below meet their expectations for manuscript improvement.
Comment 2: While the review is exceptionally well-executed, I have a few minor suggestions to enhance its pedagogical value and comprehensiveness: - I recommend adding a paragraph that summarizes the limitations of Drosophila as a model organism. This would ensure the review is 100% didactically robust and all-encompassing.
Response 2: As suggested, we added section 4.6 titled “Limits of Drosophila melanogaster when modeling human diseases” in which we analyze fruit fly limits; we also added a table (now Table 4) which also summarizes the main fly/human differences in drug metabolism. Finally, we added a specific section at the end of the Conclusions and an additional figure (now Figure 3), which illustrates the limits of this model organism.
Comment 3: It would be beneficial to include a table summarizing the various human pathologies described as potentially modellable in Drosophila. This would improve the readability and utility of this section.
Response 3: To meet the reviewer’s request, we added two tables (now Table 1 and Table 2), at the end of sections 4.1.3 (summarizing the main Drosophila cancer models) and 4.2 (summarizing Drosophila models of PACS1-PACS2-WDR37 neurodevelopmental disorders).
Comment 4: Finally, I suggest adding a few more references to the list of valuable discoveries proposed on page 3 (e.g., those from Levine, Brand, Perrimon, Rubin, etc.) to further solidify this section. Beyond these minor suggestions, I find the review to be complete and well-written. I approve its publication in IJMS.
Response 4: We are grateful to this Reviewer for this suggestion; we agree that citing more references in this section will further improve the pedagogical value of our submission. Please see the updated list of references.
Reviewer 3 Report
Comments and Suggestions for Authors
A thoroughly enjoyable walk down the long memory lane of Drosophila science. The review is well done.
Some points to consider:
On page 3, after the Drosophila Nobel prize winners, there are three paragraphs on non-fly people. The Beadle and Tatum experiments are fundamental, and Axel and Buck did amazing work, but why select these four? Why not C elegans prizes?
Comments on the Quality of English Language
There are some grammatical issues in the writing.
Author Response
Comment 1: A thoroughly enjoyable walk down the long memory lane of Drosophila science. The review is well done.
Response 1: We are very grateful to Reviewer 3 for the very positive comments, we are very happy that s/he enjoyed our manuscript.
Comment 2: Some points to consider: On page 3, after the Drosophila Nobel prize winners, there are three paragraphs on non-fly people. The Beadle and Tatum experiments are fundamental, and Axel and Buck did amazing work, but why select these four? Why not C elegans prizes?
Response 2: We are grateful to this Reviewer for noticing this. We selected those four scientists because they started their career as drosophilists, but then switched to other systems, that allowed them to win the Nobel prize. After this Reviewer’s comment, however, we realized that this might be confusing for the Reader. For this reason, we removed those lines from the text on page 3. Consequently, citing C. elegans would be off topic, now. We hope that these changes meet the Reviewers’ expectations.